

# Internal hydraulic control in the Little Belt, Denmark. Observations of flow configurations and water mass formation.

Morten Holtegaard Nielsen[1], Lars Chresten Lund-Hansen[2], and Torben Vang[2]

[1]Marine Science & Consulting, Peder Lykkes Vej 8, 4. th., DK–2300 Copenhagen S, Denmark
[2]Marine Ecology, Biological Sciences, Aarhus University, Ole Worms Allé 1, Building 1135, DK–8000 Aarhus C, Denmark

*Correspondence to*: Morten Holtegaard Nielsen (mhn@msandc.dk)

**Abstract.** Internal hydraulic control, which occurs when stratified water masses are forced through an abrupt constriction, plays an enormous role in nature on both large and regional scale with respect to dynamics, circulation and water mass formation. Despite a growing literature on this subject surprisingly few direct observations have been made that conclusively show the existence of and the circumstances related to internal hydraulic control in nature. In this study we present observations from the Little Belt, Denmark, one of three narrow straits connecting the Baltic Sea and the North Sea. The observations, comprised of primarily along-strait, detailed transects of salinity and temperature, continuous observations of flow velocity, salinity and temperature on a permanent station and numerous vertical profiles of salinity, temperature, fluorescence and flow velocity in various locations, show that internal hydraulic control is a frequently occurring phenomenon in the Little Belt. The observations, which are limited to south-going flows of approximately two-layered water masses, show that internal hydraulic control may take either of two configurations, i.e. the lower or the upper layer being the active, accelerating one. This is connected to the depth of the pycnocline on the upstream side and the topography, which is both deepening and contracting toward the narrow part of the Little Belt. The existence of two possible flow configurations is known from theoretical and laboratory studies, but has never been observed in nature and reported before, we believe. The water masses formed by the intense mixing, which is tightly connected with the presence of control, may be found far downstream of the point of control. The observations show that these particular water masses are associated with chlorophyll concentrations that are considerably higher that in adjacent water masses, showing that control has a considerable influence on the primary production and hence the ecosystem in the area.

## 1 Introduction

In the recent years there has been a growing interest in the concept of internal hydraulic control, which occurs when a flowing, stratified fluid is subject to an abrupt constriction. This interest has often been stimulated by large scale phenomena such as the exchange of water masses between different ocean basins (e.g. Bryden et al., 1994; Whitehead, 1998; Hansen et al., 2001; Girton and Sanford, 2003; Timmermans et al., 2005; Wilkenskjeld and Quadfasel, 2005), which is associated with transport of heat and salt and so may play an important role for the global thermohaline circulation (e.g. Clark et al., 2002;



Vellinga and Wood, 2002). On a regional scale, internal hydraulic control has also attracted attention, for example in connection with estuarine circulation and bottom water renewal in semi-enclosed shelf seas and biological and environmental effects thereof (e.g. Nielsen, 2001; Roman et al., 2005; Hietala et al., 2007; Gregg and Pratt, 2010).

The purpose of this study is to present and discuss observations of internal hydraulic control (in the following and where ambiguities can be ruled out just referred to as hydraulic control) made in the Little Belt, Denmark (see Fig. 1). We find this pertinent for mainly two reasons. First, the conditions under which hydraulic control is occurring in nature are often very complex, including, primarily, irregular topography or stratification, large breadth-to-length and breadth-to-depth ratios, friction, mixing and the rotation of the earth (e.g. Farmer and Denton, 1985; Nielsen, 2001; Gregg and Pratt, 2010). In a great deal of existing theoretical or laboratory studies that have lead to much of the existing knowledge on this subject some or all of these conditions have often been avoided in order to make the problem tractable. Most of these existing studies have been reviewed by Garrett (2004), Ivey (2004), Lane-Serff (2004) and Pratt (2004), all in the same journal issue. Given the complex conditions in nature it is often hard to assess the applicability of various theories and approaches, including results of numerical models (e.g. Helfrich and Pratt, 2003). Attempts at validation of existing theories and models have included comparison of calculated total flow rates with those observed (Whitehead, 1998; Borenäs and Lundberg, 2004; Pratt, 2004). However, bulk numbers would seem inappropriate for such comparisons as they may easily conceal errors in the theory or model as well as inaccuracies in the observations (Garrett, 2004).

Second, surprisingly few direct observations seem to have been made that conclusively show the existence and the circumstances related to the occurrence of hydraulic control in stratified water masses in nature. Of particular interest in this respect are the details at the point of control, which plays a crucial role by ultimately determining the conditions in either direction of the flow (e.g. Armi, 1986). Under conditions that are not fully ideal the location of the point of control has been shown to be influenced by, e.g., bottom friction and entrainment (Pratt, 1986; Gerdes et al., 2002; Nielsen et al., 2004). Existing observations of hydraulic control in a two-layered system (or with respect to the first internal mode) in a strait include Farmer and Smith (1980), Farmer and Denton (1985) and Nielsen (2001). Other observations of control with respect to the first internal mode were made by Nash and Moum (2001), who studied the stratified flow over an isolated bank on the continental shelf. Direct observations of control of higher internal modes have also been made, although these are more subtle and play a lesser role with respect to dynamics, circulation and water mass formation (e.g. Farmer and Denton, 1985; Gregg and Klymak, 2014). On the other hand, a large number of observational studies have suggested the existence of hydraulic control in various locations without being able to observe the phenomenon directly (e.g. Farmer and Armi, 1988; Armi and Farmer, 1988; Pratt et al., 2000; Fristedt et al., 2005; Girton et al., 2006; Gregg and Pratt, 2010). In some of these cases pronounced asymmetry and intense mixing around the assumed point of control are absent, suggesting that friction plays an important role in keeping the flow subcritical. In other cases the existence of a complex topography or an irregular stratification imply that the observations are very difficult to interpret from a hydraulic point of view.





Direct observations of hydraulic control under natural conditions may indeed hold the information that could be used to either validate or improve existing theories. Numerous investigations of hydraulic control reported in the literature have clearly shown a controversy regarding the principal question, namely how to determine criticality? Various approaches are based on energy considerations, calculation of the speed of long internal waves or the requirement that the flow remains well-behaved at the point of control (Farmer and Denton, 1985; Lawrence, 1993; Hogg et al., 2001). See Garrett (2004) and

references therein for a discussion of some of these issues. It seems that the various theoretical approaches lead to the same result only when the most restrictive and simplifying assumptions are applied. It is beyond the scope of the present study to pursue this issue. However, we do hope that the observations that we present here may stimulate new thoughts that could lead toward a unifying solution.

An important aspect of the Little Belt is that the topography is subject to relatively smooth variations in terms of both breadth and depth, in contrast to most of the other situations in nature in which internal hydraulic control has been observed directly (Farmer and Smith, 1980; Farmer and Denton, 1985; Nash and Moum, 2001). This implies that the Little Belt may be compared with and understood in terms of some simplified, existing theoretical and laboratory studies, such as Armi (1986) and Lawrence (1993). For the same reason we have limited ourselves to situations in which the stratification can be

approximated by two layers and where the barotropic forcing is relatively strong, driving both layers in the same direction through the topographic constraint. This points to an important property of the role played by hydraulically controlled flows in nature, i.e. that the control is determining the ratio between the flows in the two layers (in a two-fluid system) whereas the total flow is determined by the barotropic forcing and, typically, the total friction in the system. An important finding by Armi (1986) is that in case of a two-layer flow through a contraction either of the two layers may become the active,

accelerating one at the control. In Armi's laboratory experiment the solution was determined by selective withdrawal of the water masses on the downstream side of the contraction. In nature, however, the solution is much more likely to be determined by the total flow and the interface depth on the upstream side of the contraction. To our knowledge, observations in nature showing such two possible flow configurations in the same location have never been reported.

Downstream of the point of control, where the flow is supercritical, intense mixing is taking place, resulting from growing instabilities at the pycnocline (e.g. Pawlak and Armi, 2000). This mixing, which has been found to be much more efficient than in the subcritical part of the flow (Prastowo et al., 2009, and references therein), is centered around the pycnocline and tends to break down the stratification completely. As a result of the intense mixing associated with hydraulic control new water masses are formed that are distinctly different from those on the upstream side of the point of control. On a large scale

it is well-known how through or overflows at prominent topographic constraints, e.g. the Denmark Strait or the Gibraltar, lead to the formation of water masses that may influence the circulation and can be traced far away from their locations of origin (Price et al., 1993; Dickson and Brown, 1994; Price and Baringer, 1994; Girton and Sanford, 2003). On a small or





regional scale it is of interest to find out to what extent localized physical processes, especially energetic and efficient processes such as hydraulic control, are able to contribute to the formation of water masses and drive other processes

(Gargett et al., 2003; Gregg and Pratt, 2010). Previously we have suggested that hydraulic control in the Little Belt could be accounting for enhanced primary production and oxygen depletion near the bottom, observed in wide areas in either direction away from the narrow part (Lund-Hansen et al., 2008). This question is explored further here.

In the next section a general description of the Little Belt is given. Then we explain the methods that were used to make the

105 various observations, consisting of along-strait transects of temperature and salinity and vertical profiles and point measurements of temperature, salinity, fluorescence and/or current velocity from a number of locations. Then we present the observations and discuss their significance in relation to existing theoretical and laboratory studies. Then we provide a more general discussion of the findings, and finally conclusions are given.

## 2 Physical setting

The Little Belt is one of the three straits that form an important part of the transition area between the Baltic Sea and the North Sea (see Fig. 1). The Little Belt is comparatively long and narrow, its most prominent part being Snævringen, a 20 km long section in which the width is reduced to about 1 km. At either ends of Snævringen, named Tragten and Bredningen, the changes of cross-sectional area are considerable, the widths reducing by 5 km or more over distances of about 2 km. The narrow part is relatively deep, the depth ranging from 30 to 50 m, whereas in the contracting parts to the north and to the

south the depths are about 20 m or less. The water masses being forced through the narrow part of the Little Belt are thus experiencing both a reduced width and an increasing depth.

The currents through the Little Belt and the other Danish straits are much influenced by the meteorological conditions, which determine the water level variations in the North Sea and the Kattegat on the one side and in the Baltic on the other

(Jakobsen et al., 1997; Rohde, 1998). These result in flows into or out of the Baltic of up to several days of duration, during which the water masses at the upstream end are advected through the straits. For instance, the water masses in the Kattegat are normally strongly stratified with salinities ranging from about 15 to 34 (Rasmussen, 1997; Nielsen, 2005), and in connection with inflow to the Baltic these quickly reach the narrow part of the Little Belt. In the inner Danish waters tidal waves are generally much smaller than the meteorologically induced water level variations, and in the Baltic tides are

practically absent (Jacobsen, 1980). However, in some places, one of which is the narrow part of the Little Belt, tidal flows of considerable magnitude can be found (Jakobsen and Ottavi, 1997). We expect this to be due to refraction of the tidal waves in the contracting area to the north of the narrow part of the Little Belt.



Because of the little significance for the exchange of the Baltic, the Little Belt has received little attention in the scientific
literature, see primarily Jakobsen and Ottavi (1997) and Lund-Hansen and Vang (2004). Other descriptions of the conditions
in the areas between the Baltic and the North Sea may be found in Stigebrandt (1983), Gustafsson (2000), Nielsen (2001)
and references therein. Attention to the exchange through the three straits has often been motivated by inflows of dense water
masses, which increase the salinity and relieve anoxic conditions in the deep parts of the Baltic (e.g. Lass and Mohrholz,
2003). In this respect, the Little Belt normally plays as small role (Jakobsen, 1995). Other aspects related to the present study
are described and discussed in Kepp et al. (2006) and Lund-Hansen et al. (2008).

## 3 Methods

This study is based on a part of a large set of observations made in and around the narrow part of the Little Belt (Fig. 1). The
observations that we are using here were made in the following four ways and were made during April, May and June 2004.
1) Transects of salinity, temperature and density along and across the strait were made using a ScanFish from EIVA, an
undulating platform which was towed behind a ship. The ScanFish was equipped with an AROP 1000 CTD (Conductivity,
Temperature, Depth) from Geological & Marine Instrumentation (GMI). Here and elsewhere density was calculated from
salinity and temperature according to UNESCO (1981). The undulating speed of the ScanFish implied that a vertical profile
of the water column was made in a horizontal distance of roughly 1 km. 2) At a permanent station located at the Old Bridge
(Fig. 1) velocity was measured using a 300 kHz Workhorse Sentinel ADCP (Acoustic Doppler Current Profiler) from
Teledyne RD Instruments (RDI), which was placed on the seabed looking upward. The vertical velocity profile, recorded
every 30 minutes, was determined using 150 pings and a bin depth of 1 m. Salinity and temperature, also recorded every 30
minutes, were measured at five, fixed depths using a Coastlog system from GMI. Here we are using data from the depths of
5 m (the uppermost sensor), 15 m (the middle sensor) and 22 m (the lowermost sensor). The total depth at the station was
about 37 m. 3) At a few stations in the northern part of the Little Belt vertical profiles of salinity, temperature, density and
velocity were observed from a ship. The profiles of salinity, temperature and density were observed using an AROP 2000
CTD from GMI. The velocity profiles were observed using a 600 kHz Workhorse Monitor ADCP from RDI, which was
mounted over the side of the vessel looking downward, and calculated from about 100 pings and a bin depth of 0.5 m. 4) At
a number of stations throughout the Little Belt vertical profiles of salinity, temperature, density and fluorescence were
observed from a ship using an AROP 1000 CTD from GMI equipped with a standard fluorometer. The fluorescence data
were converted to concentration of chlorophyll a based on a large number of filtered water samples in which the chlorophyll
a content was determined by spectrophotometry. Other parts of the entire data set have been used and discussed by Kepp et
al. (2006) and Lund-Hansen et al. (2008).



## 4 Observations and discussion

Figure 2 shows a 50 km long transect of density along much of the Little Belt, covering the narrow part (roughly from 17 to 37 km on the length axis) as well as large parts of the contracting parts at either end. The observations were made using the ScanFish along the longer track as indicated in Fig. 1. The observations were carried out on 14 April 2004 between 10:13 and 15:04 (all times reported here are UTC), during a week-long period of weakly southward flow on the average, but with tidal flows of considerable magnitude, reaching an amplitude of about 1 m s-1 at the Old Bridge (data not shown). The transect shows a typical example of the conditions found in the Little Belt. At the northern end the water masses resemble those typically found in the Kattegat to the north, being strongly stratified and approximately two-layered, which may be attributed to a strong degree of wind mixing (Nielsen, 2005). In this case the water mass properties are ranging from a salinity and a temperature of about 18.5 and 6.7 °C, respectively, in the upper layer to about 26 and 4.6 °C in the lower layer, cf. the salinity-temperature plot shown to the right in Fig. 2. At the southern end the water masses are less and somewhat linearly stratified, in this case ranging from a salinity and a temperature of about 17 and 7.0 °C at the surface to 19 and 6.0 °C toward the bottom, i.e. of a different origin than the water masses found to the north. We attribute the weak and linear stratification in the southern part of the Little Belt to the depth being very variable, causing uneven mixing and transport due to wind and frequent restratification to take place. Typically, the water masses found in the narrow part of the Little Belt are well-mixed in the vertical, in this case showing little variation around a salinity and a temperature of 20.0 and 6.0 °C. In the present situation the well-mixed water masses found in the narrow part of the Little Belt are thus the result of mixing between the two water masses found to the north.

During the period in which the observations shown in Fig. 2 were made there was a weak inflow to the Baltic, which we also refer to as southward flow, of the strongly layered and rather saline water masses of the Kattegat. At the same time, however, the tidal currents in the narrow part were strong enough to advect the water masses forth and back. From the transect in Fig. 2, which was carried out during a peak of southward flow (data not shown), it is apparent that the two-layered structure to the north is undergoing rapid adjustment as the water masses are forced toward the narrow part of the Little Belt. Furthermore, we note that the pycnocline between the water masses to the north are located deeply and that the lower layer seems to be active and accelerating as the water masses approach the narrow part. We have previously argued that this adjustment is connected with hydraulic control and that the intense mixing of the water masses is a result thereof (Lund-Hansen et al., 2008). The observations shown in Fig. 2 are not providing new information in this respect. However, the observations to be shown in the following will shed new light on hydraulic control in this area, including details of the adjustment and the effects on the formation of water masses.

Figure 3 shows time series of velocity, salinity and temperature obtained at the fixed station at the Old Bridge (see Fig. 1). The time series cover the two periods that include the times of the observations shown in Figs. 4 through 8. The observations





show the conditions that often prevail in the narrow part of the Little Belt. There is a strong tidal signal of diurnal frequency and amplitude ranging from the 0.5 to 1 m s-1. In addition, there is a flow component that varies on a longer time scale, of a few days or so, which is primarily caused by wind-generated water level variations in the Kattegat and in the western Baltic (Jacobsen, 1980). This flow is of a considerable magnitude as well and may reach values of 1 m s-1 or more. Figure 3 also

shows that the water masses in the narrow part of the Little Belt are usually weakly stratified, despite the strong stratification often found at least in the Kattegat to the north (Nielsen, 2005). Given the magnitude of the barotropic flow we can quickly deduce that the flow in the narrow part of the Little Belt is often supercritical with respect to all internal modes. This circumstance was also pointed out by Jakobsen and Ottavi (1997). This is so even for the strongest stratification that one can imagine, which is a two-layer structure corresponding to the water masses of the Kattegat of a density difference of about 14

200 kg m$^{-3}$ (Nielsen, 2005). In a 30 m water column of layers of equal thickness this corresponds to a long-wave speed of about 1 m s$^{-1}$(e.g., Armi, 1986). As the cross-sectional area increases in either direction we can also deduce that situations of supercritical flow in the narrow part are associated with subcritical flow on the upstream side and so that the flow must often be subject to hydraulic control, the control point to be found at the upstream contracting part of the Little Belt. We may therefore conclude that our assumptions so far about the observations of hydraulically controlled flows have been correct

(Lund-Hansen et al., 2008). Further, we may point out that this finding is significantly above measurement errors and that various effects that could dampen the flow below criticality, such as friction or mixing, can be safely ruled out (Pratt,1986; Nielsen et al., 2004). On the other hand, hydraulic control in the Little Belt is an intermittent phenomenon that is limited to parts of the tidal cycle if the barotropic flow is weak, which is the case in some of the periods shown in Fig. 3. The length scales of the contracting parts are short enough that the hydraulic adjustment is well toward the quasi-steady limit (Helfrich,

1995).

Another along-strait transect, obtained using the ScanFish as described above, is shown in Fig. 4. It covers the contracting part to the north and some of the narrow part, as shown by the line and circles in the detailed inset in Fig. 1. The transect was made on 16 June 2004 from 12:22 to 13:37 at the time of which the flow was southward through the Little Belt, the depth-

215 averaged flow being well above 1 m s$^{-1}$ for a period of about 24 hours, cf. Fig. 3. At the upstream end of the transect the water column is approximately two-layered, the salinity and the temperature being about 22 and 13.5 °C in the upper layer and 29 and 7.5 °C in the lower layer. Toward the narrow part of the Little Belt the water column is undergoing rapid adjustment, the pycnocline dropping toward the bottom from a depth of less than 10 m over a distance of a few kilometres. In the narrow part of the strait the water masses are almost well-mixed, the density varying between 1017 and 1019.5 kg m$^{-3}$.

From the salinity-temperature plot it is seen that these water masses are the result of mixing between the two water masses on the upstream side and that the mixing is primarily the result of entrainment into the lower layer, this being diluted considerably. From these circumstances we may safely conclude that the flow is subject to hydraulic control and that the lower layer is the active one. Without accompanying observations of flow velocity we cannot say where the point of control

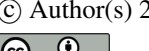



is located though. Similar observations made some hours earlier on the same day along roughly the same track can be found

in Lund-Hansen et al. (2008).

Just before the along-strait transect in Fig. 4 was made the two cross-strait transects of density shown in Fig. 5 were carried out. They both show some density variation across the strait, which could make it difficult to interpret the density structure in the along-strait transect. However, the cross-strait transects confirm that the water column is undergoing considerable

adjustment as the narrow part of the Little Belt is approached. The two cross-strait transects along with the map in Fig. 1 show some important details of the topography. One is that following the deep, central part there is little change of width, whereas the bottom depth is increasing considerably. Another is that following the upper part the width is decreasing considerably as the narrow part of the Little Belt is approached. In a situation of two layers as considered here we may thus assume the following. In case of a relatively deep pycnocline the lower layer will primarily experience a drop of the bottom,

suggesting that the lower layer becomes the active one and the flow resembling that of spilling down a slope (e.g., Pawlak and Armi, 2000). In that case one would expect to find the point of control near the top of slope. On the other hand, in case of relatively shallow pycnocline the upper layer will be much influenced by the decreasing width. This suggests that the upper layer becomes the active one, that the flow resembles that through a contraction and that the point of control is to be found near the narrowest part of the contraction (e.g. Armi, 1986). In nature hydraulic control are also influenced by other

circumstances, e.g., friction or mixing, both of which tend to move the control point in the downstream direction (Pratt, 1986; Nielsen et al., 2004).

The vertical profiles of density and velocity shown in Fig. 6 were recorded in a situation much like the one in Figs. 4 and 5. The observations were obtained on 11 May around 19:00 during which the flow was southward of a speed of about 1 m s$^{-1}$ at

the Old Bridge, cf. Fig. 3. These observations were made from a ship using a CTD that was profiled vertically through the water column and an ADCP that was mounted over the side looking downward. The profiles were recorded at two stations near what is assumed to be the location of the control point. The observations show a roughly two-layered water column with a density difference of about 8 kg m$^{-3}$. At the upstream station the pycnocline was located at a depth of about 15 m, i.e. somewhat deeper than what was found in Figs. 4 and 5, and the velocity ranged from 0.4 to 0.6 m s$^{-1}$ with the upper layer

moving slightly faster. At the other station, located about 2 km in the downstream direction, the pycnocline had dropped to a depth of about 30 m, the upper layer velocity was still around 0.6 m s$^{-1}$, but the lower layer velocity had increased to about 1.2 m s$^{-1}$. We note that there were significant secondary currents, which we attribute to the complexity of the topography.

The combination of observations of stratification and velocity in this situation allows us to calculate the composite Froude

number. Since the relative density, $\delta = (\rho_2 - \rho_1)/\rho_2$, between the layers is small the composite Froude number can be approximated by



$$G^2 = F_1^2 + F_2^2 \quad ,$$

where $F_i^2 = U_i^2 / \delta g y_i$ , $y$ and $U$ refer to the layer depth and velocity respectively, and indices 1 and 2 refer to the upper and the lower layer respectively (Armi, 1986). Taking layer depths of approximately 15 and 15 m at Station G53 and 30 and 15 m at Station G51, we find composite Froude numbers of $G^2 = 0.31 + 0.14 = 0.45$ at G53 and $G^2 = 0.16 + 1.25 = 1.41$ at G51. The fact that the downstream value is considerably above unity and that the area between the two stations is associated with abrupt changes of layer depths and velocities as well as the level of the pycnocline can be taken as solid evidence that this situation is subject to hydraulic control. This fact is not changed much if deviations from the two-layer assumption were to be taken into consideration, e.g. by calculating the effects of the density and the velocity profiles, cf. Garrett (2004). The observations further show that in this situation, the lower layer being the active one and the flow spilling down the slope, the point of control is located somewhere between Station G53 and G51. This location is a bit downstream of the top of the slope, suggesting that friction and/or entrainment play a significant role in determining the control (Pratt, 1986; Nielsen et al., 2004).

Figures 7 and 8 show a number of vertical profiles of density and chlorophyll a along with the corresponding distributions of salinity and temperature obtained on two separate occasions on stations covering most of the Little Belt, cf. Fig. 1. Both situations are concerned with inflow to the Baltic, and in both cases the water masses on the upstream side are strongly stratified and approximately two-layered, as seen in the top rows of the two figures. An important difference between the two situations is related to the depth of the pycnocline on the upstream side. In the first situation the pycnocline is relatively deep, and in the other it is relatively shallow, which is seen to result in a difference in the hydraulic response between the two situations. The observations shown in Fig. 7 were made on 4 and 5 May 2004 during a weak southward flow on the average, but at the same time the tidal flows were considerable, the tidal amplitude being about 0.5 m s$^{-1}$, cf. Fig. 3. On the northern stations, shown in the upper panel in Fig. 7, the two water masses are characterized by a salinity and a temperature of about 17 and 10 °C in the upper layer and about 30 and 5 °C in the lower layer, and the pycnocline is found at a depth of 12-15 m. In the narrow part of the Little Belt, seen in the middle row, the water masses are still mostly two-layered, and the upper layer is having roughly the same properties as found on the upstream side. However, in the lower layer a strong entrainment of upper layer water has taken place, resulting in a salinity and a temperature of about 24 and 7 °C at Station 20. At the same time the pycnocline has dropped to almost 20 m, strongly indicating that the flow is hydraulically controlled and that the lower layer is the active, accelerating one. At Station 21, south of the narrow part and seen in the lower row in Fig. 7, water masses of the same properties as produced by the strong entrainment in the narrow part are found. Much further to the south the density structure is almost the same, but the water masses are much different, showing that in this situation the mixed water masses resulting from hydraulic control in the narrow part are primarily found locally around the contraction.





The observations shown in Fig. 8 were made on 18 and 19 May 2004 during a period of moderate inflow to the Baltic and
tidal flows of an amplitude of about 0.5 m s$^{-1}$, which was not enough to reverse the overall flow (see Fig. 3). Notably, the
pycnocline to the north is rather shallow, at a depth of 6-8 m, and the water masses are characterized by a salinity and a
temperature of 18 and 12 °C in the upper layer and about 30 and 6 °C in the lower layer. In the narrow part of the Little Belt
the lower layer is only modified slightly, but in the upper layer a strong entrainment of lower layer water appear to have
taken place. Furthermore, the density structure of the upper layer resembles that of a supercritical flow (e.g. Pawlak and
Armi, 2000), all of which indicates that the flow is subject to hydraulic control and that the upper layer is the active,
accelerating one in this situation. In fact, the observations made at the permanent station at the Old Bridge (Fig. 3), which is
located about midway between Stations 51 and 20, show that the upper part of the water column is indeed moving
considerably faster than both the average and the lower part. Thus, compared to the situation shown in Fig. 7 a different flow
configuration is found. The cross-strait transects shown in Fig. 5 suggest that in case of a shallow pycnocline the upper layer
is subject to an abrupt change of width when approaching the narrow part of the Little Belt, which is probably decisive for
the flow configuration. Again, the water masses formed by hydraulic control are found locally downstream of the narrow
part (at Station 21), whereas further to the south the water masses are much different. However, in this situation some of the
water masses formed in the narrow part are found near the bottom at Station 27, some 30 km to the south of the narrow part.
The observations shown in Figs. 7 and 8 were not accompanied by observations of flow velocity. Therefore we cannot say
where the points of control for the two flow configurations are located.

An interesting picture emerges when considering the distributions of chlorophyll a in relation to the different water masses
found in the two situations shown in Figs. 7 and 8. First, it is noticed that away from the narrow part of the Little Belt the
concentration of chlorophyll a is having a maximum at some depth that is coinciding with the density gradient, and above
and below it is considerably lower. This is clearly seen in both the upper and the lower panels of Figs. 7 and 8, showing the
conditions on the upstream side and on the downstream side of the narrow part of the Little Belt, respectively. Such
subsurface maxima are a prominent feature in areas that are strongly stratified and in which the growth of phytoplankton in
the surface mixed layer is dependent on mixing across the pycnocline (e.g. Lyngsgaard et al., 2014). In contrast to this, at the
stations where the water masses formed in connection with hydraulic control are found, both in the narrow part of the Little
Belt (the middle panels) and on the downstream side (the lower panels), it is seen how these particular water masses are
associated with markedly higher concentrations of chlorophyll a. In addition, these high concentrations are no longer
confined to areas deep in the water column, but are extending toward the surface. Taking the increased chlorophyll a
concentration as a proxy for primary production and increased phytoplankton concentration, it would seem that when the
mixing associated with hydraulic control is bringing inorganic nutrients into the euphotic, upper part of the water column,
the result is a quick response of the phytoplankton in terms of increased primary production and growth.





Previously, we have shown that the Little Belt is associated with locally high depth-integrated primary production and have suggested that this was closely related to mixing due to hydraulic control (Lund-Hansen et al., 2008). The observations in Figs. 7 and 8 clearly show that the water masses formed in connection with hydraulic control are indeed the ones associated

with increased primary production and growth of phytoplankton. These water masses may be found locally around the narrow part of the Little Belt, but also some distance away. The latter is seen at Station 27 in Fig. 8, where the water masses formed in connection with hydraulic control are found near the bottom. Despite their deep location where little light is available these water masses contain notably more chlorophyll a than at Station 29 further to the south. Therefore we may conclude that our suggestion was correct.

**5 Discussion**

In the present study we have limited our investigations in the Little Belt to a small number of situations of two-layered, southward flow. We have observed the existence of hydraulic control in these situations with certainty. In addition we have observed that the topography and the depth of the pycnocline play important roles in determining the flow configuration. Despite the limited number of observations in this study, the magnitude of the barotropic flow along with the stratification

found to the north or to the south show that hydraulic control in one form or another occurs on a regular basis. Most certainly this includes control also of higher internal modes and, necessarily, so-called approach control (Farmer and Denton, 1985; Armi, 1986). Further, more detailed observations will have to shed light on these circumstances.

When contemplating the significance for a system consisting of two basins connected by a strait it is fundamental to realize

the role that hydraulic control is playing. In oceanic or estuarine flows the total exchange of water masses through the strait is usually determined by the water level difference between the basins, which equal the head loss along the strait. In a two-layer flow the effect of hydraulic control is to determine at what proportion the total flow is divided between the flows of the different water masses. A good example is Øresund, the second largest of the three straits in the transition area between the Baltic Sea and the North Sea, in which the total flow is mainly determined by the frictional loss at the shallow and wide

Drogden Sill (Jakobsen et al., 1997). In connection with southward flow hydraulic control in the contraction in the deep, northern part of Øresund is determining the proportion of upper and lower layer water of the Kattegat that can pass the Drogden Sill and flow into the Baltic (Nielsen, 2001). Knowing the relation between the stratification and the flow at the point of control, i.e., a so-called weir formula (Pratt, 2004), one could ideally determine the variables of the flow as well as the dynamical properties of the system. In the Little Belt a similar functionality exists.

In the present study we have found that during periods of two-layered flow, which occurs often due to the stratification in the Kattegat (Nielsen, 2005), hydraulic control in the northern part of the Little Belt may be associated with two different flow configurations, i.e. the active, accelerating layer being either the upper or the lower one. Which of the two flow




configurations actually occurs is a function of the depth of the pycnocline on the upstream side and probably also the total

355 flow (Armi, 1986). This is so because the upper and the lower layer are primarily influenced by the reduction in width and the slope of the deep, central part, respectively. Different configurations of the flow through a contraction are known from the laboratory experiments by Armi (1986). In nature one should also expect that locations exist where different flow configurations may occur. However, we believe that this is the first time that this has been observed and reported in the scientific literature.

The observations have shown that the flow configuration seems to have a strong effect on the water masses that are being formed in connection with hydraulic control. When the lower layer is the active one, entrainment of upper layer water results in water masses of relatively low salinity. On the other hand, when the upper layer is the active one, entrainment of lower layer water leads to water masses that are relatively saline. In fact, the situation shown in Fig. 8 was associated with inflow

to the Baltic Sea of the highest salinities in 2004 (data not shown). In addition, when considering the influence that hydraulic control might have on the conditions in the adjacent areas the amount of the formed water masses is important. This is a function of the ratio of flow in the two layers allowed by hydraulic control and the magnitude and the duration of the barotropic flow. In the two situations shown in Figs. 7 and 8 the first was associated with a relatively weak barotropic flow and had had a short duration at the time of observation. In contrast, the second was both stronger and of longer duration and

was associated with formation of water masses that were found far to the south of the narrow part of the Little Belt. Intermittency of the tidal flows makes the picture a little unclear, but due to its relatively small volume the narrow part of the Little Belt is quickly flushed and so is mostly reflecting the inflowing water masses at any given time.

Regardless of the flow configuration the observations reported here point to the fact that hydraulic control and the associated

mixing in a strait is a mechanism that forms new, distinct water masses that may influence the circulation and other processes in the adjacent areas. In this case this is probably the reason why the Little Belt is generally associated with a higher primary production, higher concentrations of phytoplankton as well as recurrent oxygen depletion near the bottom (Jørgensen and Richardson, 1996; Lund-Hansen et al., 2008). Biological processes and the ecosystem dynamics as a whole are often driven by physical processes (Legendre and Demers, 1984). There may well be a great number of estuarine areas

elsewhere in the world the functioning of which cannot be understood properly in the light of other prevailing physical processes, e.g. wind mixing, and where hydraulic control could play a decisive role (Roman et al., 2005). Again, knowing the relation between stratification and flow at the critical point, hydraulic control could potentially be used to quantify the transport of water masses and the mixing between them and to determine the influence on other processes. The controversy regarding how to determine criticality suggests that some comprehension and much work are needed before hydraulic

control in nature can be utilized in this manner (Garrett, 2004).



As discussed above we expect the stratification in the southwestern Kattegat to have a strong influence on the effect that hydraulic control has on the inflow of water masses to the Baltic Sea. During the winter time the pycnocline is usually deeply located, due to strong wind mixing in the Kattegat (Rasmussen, 1997). In fact, the pycnocline may be found at such a

depth that stratification is absent and that internal dynamics is not occurring in the Little Belt at all. During the summer, on the other hand, the pycnocline may be relatively shallow, due to little wind mixing and heat input from the atmosphere. In addition, local winds may create up or downwelling. The combination of baroclinic and barotropic forcing, which are varying somewhat independently, implies that hydraulic control in the Little Belt and its influence on the formation of water masses, inflow to the Baltic, primary production and ecosystem dynamics is a complex and highly dynamic process. The full

effect of this can probably only be examined and understood using an adequate model.

## 6 Conclusions

This observational study has shown that internal hydraulic control is a frequently occurring phenomenon in the Little Belt. The observations add to a surprisingly small number of studies that have conclusively shown the existence of control in stratified flows in nature. We have limited our observations to situations of south-going flow in which the water masses on

the upstream side are approximately two-layered, facilitating a comparison with some existing laboratory and theoretical studies. As the layered water masses approach the narrow part of the Little Belt they are subject to both a deepening and a narrowing topography. The two layers undergo a rapid hydraulic adjustment with one of the layers accelerating strongly and intense mixing around the pycnocline taking place. In one situation vertical profiles of salinity, temperature and flow velocity show a considerable increase in the composite Froude number over a short distance of the flow. In this situation the

lower layer is the active, accelerating one, the flow is spilling down the slope in the middle of the strait, and the point of control is found to lie a bit downstream from the top of the slope. In another situation in which the pycnocline on the upstream side is relatively shallow we observe the upper layer to be the active, accelerating one. In this situation the point of control is presumably located near the narrowest part of the upper, contracting part of the topography. Such two different flow configurations, which are known from theory and experiments in the laboratory, have never been observed in nature

and reported before, we believe. Despite the limited number of observations the magnitude of the barotropic, partly tidally driven flows implies that internally critical flows are a recurring phenomenon under most circumstances in the Little Belt. Certainly this includes control of higher internal modes and approach-controlled flows.

The observations show that internal hydraulic control in the Little Belt is associated with intense mixing and formation of

large amounts of water masses, which can found far away on the downstream side of the control. These water masses are subject to much higher levels of chlorophyll a than the surrounding water masses, substantiating a previous study of ours that increased primary production in the adjacent areas is strongly influenced by control in the narrow part of the Little Belt. The two different flow configurations have a strong effect on the formation of water masses, the active, accelerating layer being



subject to intense entrainment and dilution. Thus, as the upper layer was observed to be the active one, water masses of the
420 highest salinities in 2004 were produced and brought into the Baltic.

**Author contributions**

All authors took part in collecting the data. M. H. Nielsen carried out the data analysis and prepared the manuscript with
contributions from the co-authors.

**Acknowledgements**

The map in Fig. 1 is based on the Area Information System – Bathymetry for the Inner Danish Waters, copyright by the
Danish Ministry of Environment. The observations shown in Figs. 2, 4 and 5 were made from R/V Tyra. The observations
shown in Fig. 6 were made from R/V Genetica II. We are thankful to the crews of these vessels for their help and support.
The data shown in Fig. 3 were obtained by the County of Vejle, and the data shown in Figs. 7 and 8 were obtained by the
County of Funen. These data were made available through the Danish Natural Environment Portal. This study received some
430 financial support from the Danish Natural Science Research Council (grant no. 4344-6757654).

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

**Figure captions**

Figure 1. Map of the Little Belt with insets showing the three straits connecting the Baltic and the North Sea (upper left) and a detailed view of the narrow part and the contracting part to the north (right). The depth contours indicate 10 m (solid), 20 m (dashed) and 30 m (dotted). The original place names that are briefly mentioned in the text are shown. The locations of the stations from which data have been used are shown (squares), OB implying the Old Bridge. The two along-strait survey lines, along which every 2 and 5 km are marked (circles), are where the density transects in Figs. 2 and 4 were made. The two cross-strait survey lines just north of the narrow part are where the density transects in Fig. 5 were made.

Figure 2. Along-strait transect of observations of density (minus 1000 kg m⁻³) given by contour lines (left panel) and the corresponding distribution of salinity and temperature (right panel). In the salinity-temperature plot the lines are contours of constant density (minus 1000 kg m⁻³). The observations were made using a ScanFish (see text), the entire transect corresponding to more than 40 undulations from surface to bottom. The transect was made on 14 April 2004 between 09:10 and 15:07 along the long survey line shown Fig. 1, the length of the transect being measured from the northern end.





Figure 3. Time series of observations from the station at the Old Bridge (see Figure 1) for 2 through 19 May and 14 through 17 June 2004. The upper panel shows velocity measurements projected onto the main flow direction, positive values for northward flow through the Little Belt. The curves are the depth-averaged value (solid, thick), the bin box at a depth of about 550 5 m (solid, thin), and the bin box at 3 m above the bottom (dashed, thin). The total depth at the station is about 37 m. The middle and the lower panel show the salinity and the temperature, respectively, measured at a depth of 5 m (solid, thin curve), 15 m (solid, thick curve) and 22 m (dashed, thin curve).

Figure 4. Along-strait transect of observations of density (minus 1000 kg m$^{-3}$) given by contour lines (left panel) and the 555 corresponding distribution of salinity and temperature (right panel). In the salinity-temperature plot the lines are contours of constant density (minus 1000 kg m$^{-3}$). The observations were made using a ScanFish (see text), the entire transect corresponding to 15 undulations. The transect was made on 16 June 2004 from 12:22 to 13:37 along the short survey line as shown Fig. 1.

Figure 5. Cross-strait transects of observations of density (minus 1000 kg m$^{-3}$) given by contour lines. The transects were 560 made using a ScanFish (see text) along the two cross-strait survey lines just north of the narrow part of the Little Belt as shown in Fig. 1. The transects were made on 16 June 2004 from 09:45 to 10:56. The northern transect is shown in the left panel, the southern in the right, and in both cases the length axes are pointing from north-west to south-east.

Figure 6. Vertical profiles of density (thick lines) and velocity (thin lines) measured at the two stations G53 (left panel) and 565 G51 (right panel), shown in Fig. 1. The observations were made on 11 May 2004 at 19:10 and 18:20 respectively. The velocity is projected onto the main flow direction (solid line; positive southward through the Little Belt) and perpendicular to this (dashed line; positive to the right of the main flow direction).

Figure 7. Vertical profiles of density (panels to the left) and chlorophyll a (panels in the middle), and the corresponding 570 distributions of temperature and salinity (panels to the right). In the salinity-temperature plot the lines are contours of constant density (minus 1000 kg m$^{-3}$). The observations were made on 4 May 2004 between 07:35 and 08:30 at stations 14 (circles), 16 (squares) and 18 (crosses), all shown in the upper row, on 4 May at 08:51 at station 51 (circles) and on 5 May at 05:17 at station 20 (squares), shown in the middle row, and on 5 May between 05:45 and 08:49 at stations 21 (circles), 27 575 (squares) and 29 (crosses), shown in the lower row.

Figure 8. Vertical profiles of density (panels to the left) and chlorophyll a (panels in the middle), and the corresponding distributions of temperature and salinity (panels to the right). In the salinity-temperature plot the lines are contours of constant density (minus 1000 kg m$^{-3}$). The observations were made on 18 May 2004 between 08:19 and 09:16 at stations 14



(circles), 16 (squares) and 18 (crosses), all shown in the upper panel, on 18 May 09:34 at station 51 (circles) and on 19 May 05:26 at station 20 (squares), shown in the middle panel, and on 19 May between 05:52 and 08:41 at stations 21 (circles), 27 (squares) and 29 (crosses), shown in the lower row.



Figure 1







Figure 2

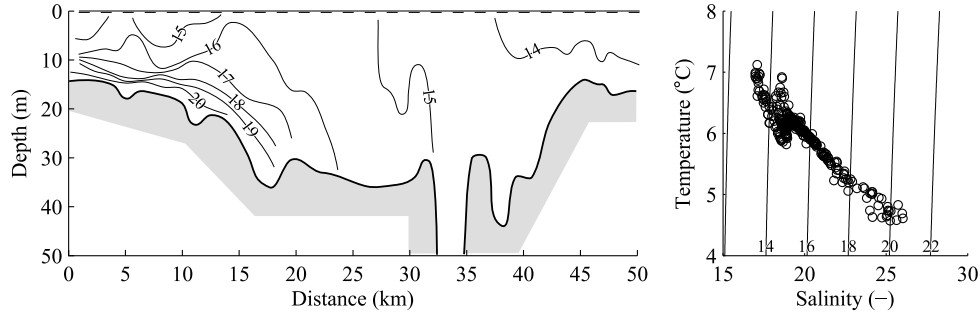





Figure 3

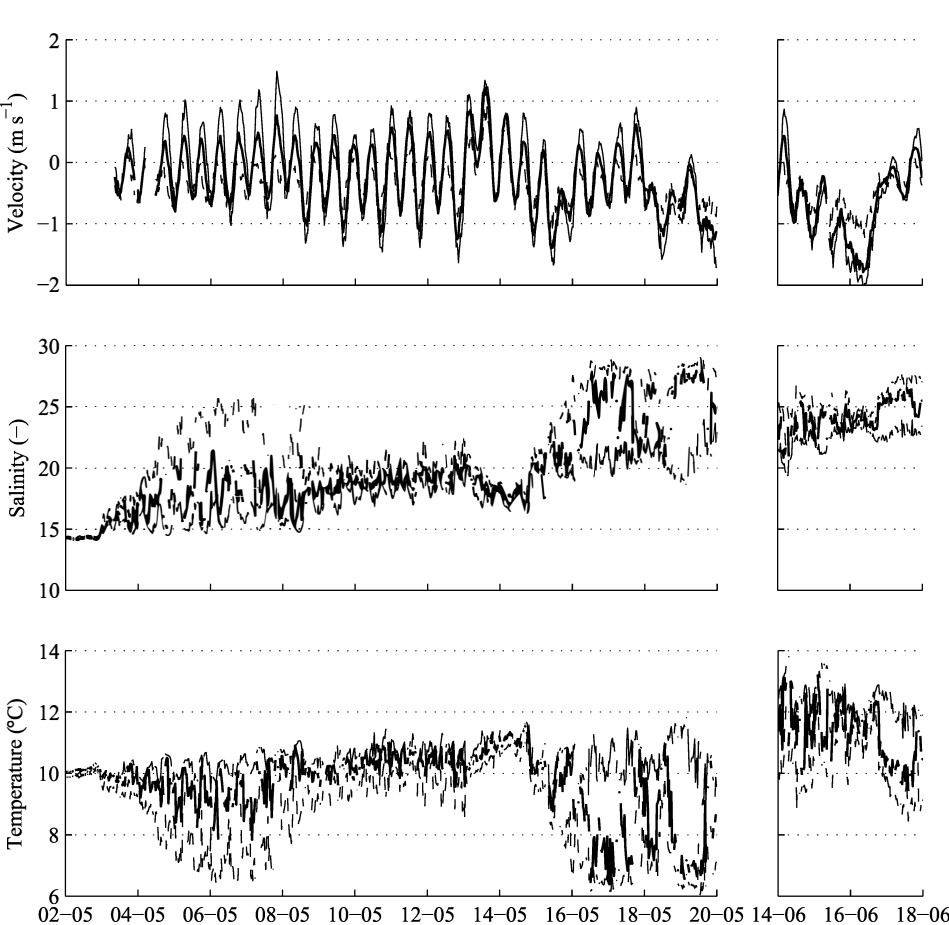




Figure 4

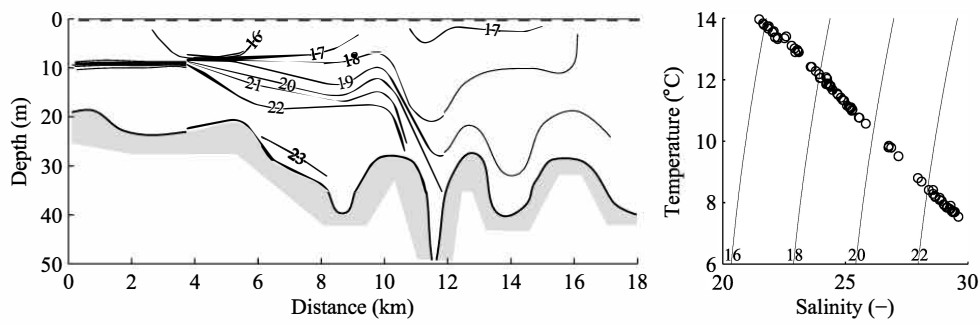




Figure 5

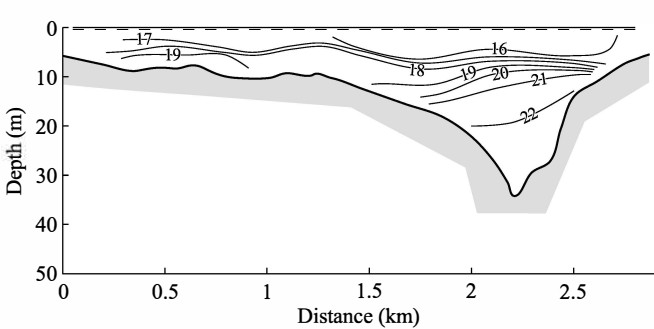
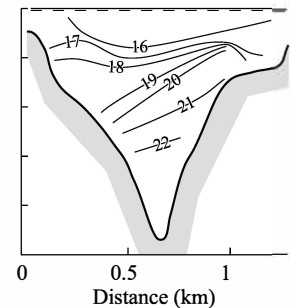





Figure 6

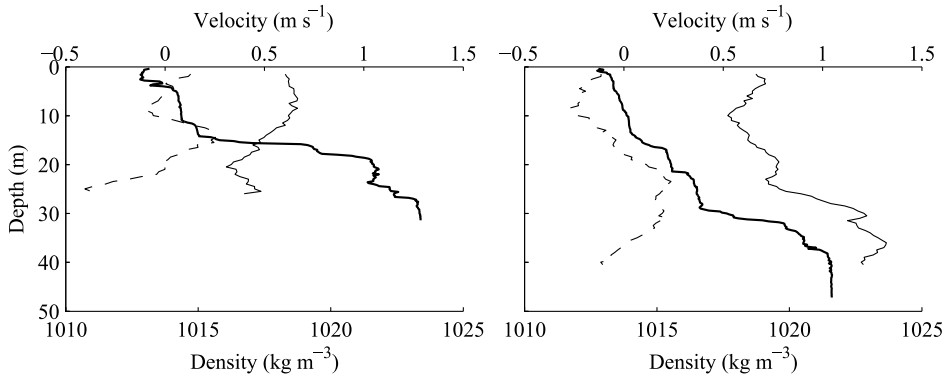





Figure 7

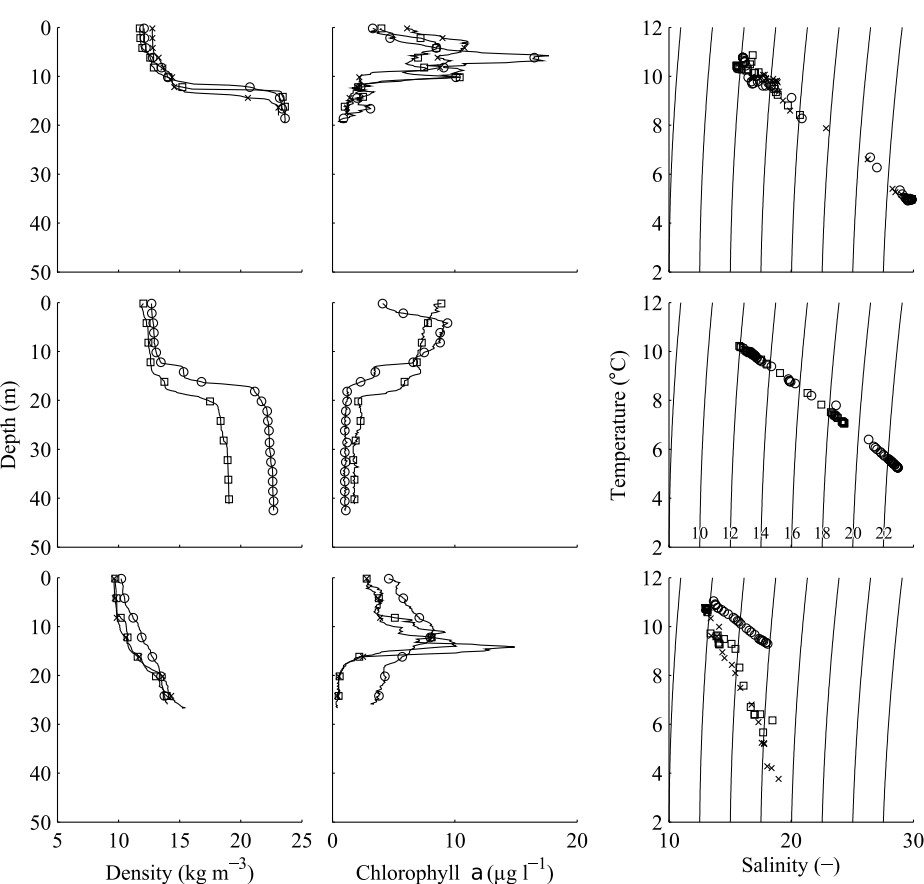


Figure 8

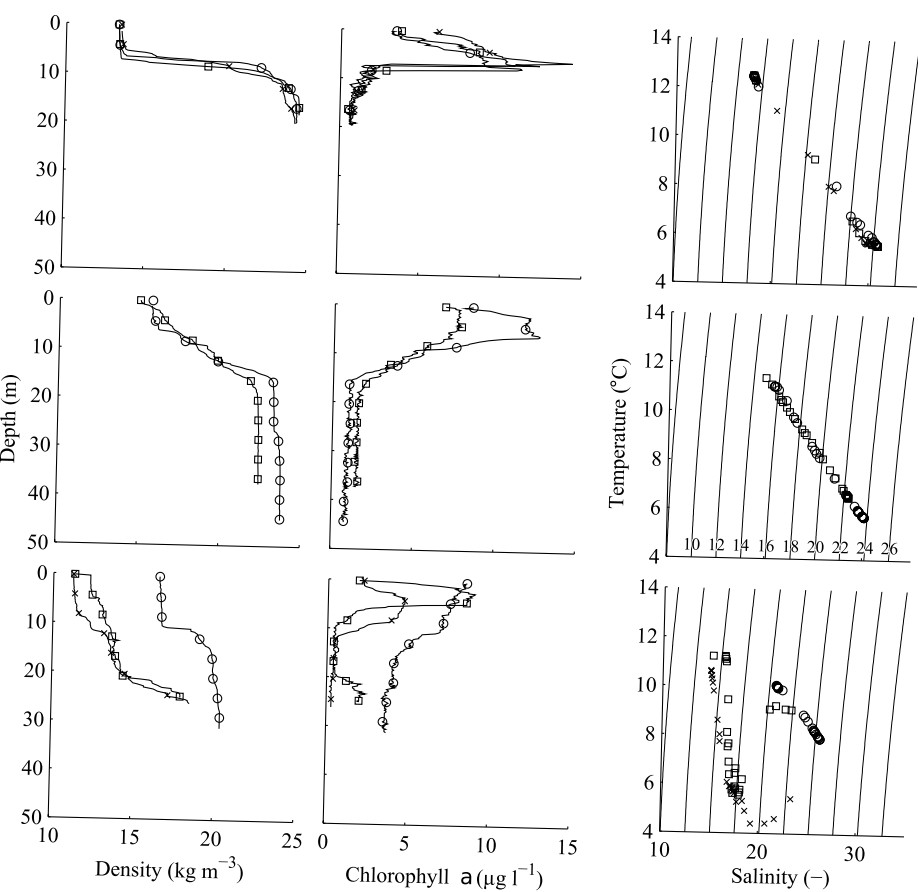