# Peer review of "Internal hydraulic control in the Little Belt, Denmark. Observations of flow configurations and water mass formation."

_Ocean Science, 2017_

## Referee Comment (RC1) · Anonymous Referee #1 · 23 May 2017

**General comments:**

This is a presentation of an interesting data set in Little Belt that shows a transition from internally subcritical to supercritical flow as the water enters a narrow and deep part of the strait. Further downstream, the water is much less stratified, indicative of large mixing taking place in the supercritical flow, and fluerescence data also indicate enhanced primary productivity due to mixing of nutrients into the euphotic zone. The data clearly shows situations where the lower layer accelerates down the slope and becomes the active layer. There are also indications of a situation where the upper layer accelerates, thins, and becomes active due to the decreasing width of the channel. I think this is an interesting work that deserves publication. The presentation is

brief, well structured, and mostly clear. However, I have some issues which I would like adressed before publication:

Specific comments:

1. The discussion is rather qualitative, except for the estimate of Froude numbers. Particularly when it comes to mixing, I would like a more quantitative discussion of the claimed water transformations, and the reasonability of those. One way to help the reader would be to show the TS diagrams both upstream and downstream of the transformation region. Another would be to estimate the volume fluxes of incoming and transformed waters. For example, in Figure 2, the downstream water is of nearly the same density as the upper layer upstream. Is it reasonable that the upstream volume fluxes combine into this light water mass, or has the dense water simply not reached the downstream section yet? The same is the case in Fig. 4, and there I would like some further arguments that the sharp front is a stationary control rather than a propagating bore. An energy argumentation would also help. Is the loss of kinetic energy sufficient to explain the claimed rise in potential energy due to mixing? I think it is worth putting some effort into this, since this is one of the most extreme cases I have seen regarding mixing and removal of stratification downstream of a jump. That should also be one of the main key points of the manuscript.

2. I miss a discussion about the importance of the curvature of the strait. The strait is dramatically meandering, and transects upstream of the most curving sections show strong interface tilting. How much can the interface be expected to tilt in the curving parts, and how does that influence the along-strait transects? May it also influence the hydraulics, the mixing, and the friction in the strait?

3. In the claimed scarcity of published observations of controlled flows the authors miss a number of high-quality observations over fjord sills that show the controlls and associated mixing in such flow configurations in much more detail than what is presented here, e.g. Farmer and Armi (1999), Klymak and Gregg (2004), Inall et al. (2004),
Staalstrom et al. (2015).

Detailed comments:

Line 22: "that" should be "than".

Paragraph starting at line 48: Include references to high resolution fjord sill studies.

Lines 91-93: How much larger is the mixing efficiency in supercritical flow? I find no such conclusion in the referred paper. I would also like some references where the stratification breaks down totally after a control. Most of those I know about maintain a stratification, although modified, downstream of the control.

Line 113: "reducing" should be "changing" since reducing is only applicable in one end.

Line 126: "refraction" is the change of wave propagation direction due to change in propagation velocity, which is not what happens here.

Line 142: "vertical profile" is misleading since the probe does not move vertically.

Line 168: "less and somewhat linearly stratified". Unclear.

Figures 2 and 4: Show TS data both upstream and downstream. Also indicate in the transect where the TS diagrams have been taken.

Figure 3: Please indicate the timing of the various transects.

Lines 208-210: How short are the length scales and how short should they be for the flow to be quasi-steady?

Lines 221-222: Please add a quantitative discussion about the reasonability of this. Is it sure that the front is not moving?

Lines 229-230: What happens downstream when the curvature is much more important?

Lines 261-263: Strictly speaking what is shown is that the flow goes from subcritical to
supercritical. You do not show where the critical cross-section is, and that it is steady in time. I.e. you do not show that the flow is controlled.

Lines 292-295: Again, argue more quantitatively. The evidence of a control is quite weak.

Lines 314-320: How does the increased primary production in the upper layer fit with the earlier conclusion that entrainment is mainly into the bottom layer?

Lines 326-327: Again this is somewhat confusing. Entrainment is mainly into the upper layer, but downstream the upper layer is at the bottom. Some more explanation needed to make the text clear.

Line 340: In estuarine flows the exchange is often determined by internal hydraulics rather than sea level differences.

Lines 371-372: Again a quantitative discussion is needed. How quickly is Little Belt flushed out?

Lines 381-383: How can information about the critical point be used to quantify mixing?

Line 412: Since such phenomena have not been observed, one could include "the possibility" to avoid stating that they exist in reality without evidence.

OSD

---

## Referee Comment (RC2) · Anonymous Referee #2 · 5 Jun 2017

The manuscript "Internal hydraulic control in the Little Belt, Denmark. Observations of flow configurations and water mass formation" by Nielsen et al. investigates observations in the northern part of the Little Belt during two-layered, southward flow and connect them to hydraulic control. Two configurations are described, where either the upper or the lower layer is the active and accelerating one. These configurations are dependent on the depth of the pycnocline on the upstream side. The finding of this study are interesting and worth publishing, after following comments are addressed.

Line 25: Although many readers may be familiar with the concept of hydraulic control, it would be useful to give a short definition (with reference) of this concept in the

introduction as it is the main topic of this study.

Line 48-49: please include references to these studies.

Line 119ff: Baltic → Baltic Sea (several times)

Method section: It would be great to have all observations used in this study shortly described in form of a table including information about location (refer to figure 1 using the station/transect numbers), the observation period, what was observed and how, and important notes (e.g. that there was an inflow during the observation period).

The name of section 3 (Methods) should be renamed to "Observations" and the name of section 4 (Observation and discussion) to "Results"

How do other processes, for example wind mixing, may have impacted the observed signals and their interpretation?

Line 205-206: please explain in more detail why effects, such as mixing can be ruled out.

Line 209: please quantify "short enough"

Line 229: please discuss the "considerable adjustment" in more detailed

Line 233ff: the definition of the two layer system in the cross-strait transects in Figure 5 is somehow unclear. Which case, described here, is assigned to the situation in Figure 5? Please clarify.

Line 260ff: The computed Froude numbers indicate a change from subcritical to supercritical flow. Please discuss in more detail how this finding gives evidence that the situation is subject to hydraulic control.

Line 267-268: How is the location of the point of control be influenced by friction/entrainment?

Line 310ff: The description of the distributions of chlorophyll a is somehow confusing.

Please first describe Figure 7 and then Figure 8.

Line 381: How can hydraulic control be used to quantify the transport of water masses, the mixing between them and also the influence of other processes?

Figure 1: To more easily identify the position of the stations, it would be great if the station positions could be more highlighted and the transects should be given a numbering. The depth contours are difficult to distinguish.

Figure 2: In the title it is written that the length of the transect being measured from the northern end (assuming distance=0), but looking at the figure itself and the text, it looks like the approximately two layered water masses are located at 50 km distance. Please include in the figure, where north and south is or the starting/ending coordinates.

Figure 3 The different curves are difficult to distinguish and need to be clarified

All figures showing salinity: The unit is missing everywhere, is it g/kg?

———————————————

---

## Referee Comment (RC3) · Anonymous Referee #3 · 9 Jun 2017

This paper on internal hydraulic control in the Little Belt by Nielsen et al. is of considerable interest. The introduction comprises an overview of the present state of multi-layer hydraulics and sections 2 and 3 (physical setting, methods) provide adequate descriptions of the location and fieldwork. My main comments pertain to section 4 dealing with the observations and discussion. First some minor points:

line 179: generally the expression used is "back and forth", but the authors may possibly have sopme ingeious reason for reversing the standard order.

line 181: pycnocline (singular) is located

line 199: for heightened clarity write ...  the upper and lower water masses of the

[Figure]

Kattegat with a density difference...

These were minor matters but my next comment is more important: line 228: It is stated that the cross-strait density transects showed density variations across the strait, cf. fig. 5. An examination of in particular the right panel of fig 5 reveals a slope of such a magnitude that a geostrophic balance is a distinct possibility, and hence that it is doubtful whether the system can be regarded as nonrotational and farther down the page line 255 et seq. the authors apply Armi's nonrotating formalism. It is necessary to include a discussion of why rotational effects can be neglected!!

Sections 5 and 6 (discussion and conclusion) are satisfactory and I particularly appreciated that the authors invoked the hydraulic effects on the biogeochemical processes, something I do not think has been done before.

A minor point here is that on line 390 ... internal dynamics (plural) are not. . .

---

## Author Comment (AC1) · 12 Jul 2017

COMMENTS FROM REFEREE: This is a presentation of an interesting data set in Little Belt that shows a transition from internally subcritical to supercritical flow as the water enters a narrow and deep part of the strait. Further downstream, the water is much less stratified, indicative of large mixing taking place in the supercritical flow, and fluerescence data also indicate enhanced primary productivity due to mixing of nutri-ents into the euphotic zone. The data clearly shows situations where the lower layer accelerates down the slope and becomes the active layer. There are also indications of a situation where the upper layer accelerates, thins, and becomes active due to the

decreasing width of the chan- nel. I think this is an interesting work that deserves publication. The presentation is brief, well structured, and mostly clear. However, I have some issues which I would like adressed before publication: Specific comments: 1. The discussion is rather qualitative, except for the estimate of Froude numbers. Particularly when it comes to mixing, I would like a more quantitative discussion of the claimed water transformations, and the reasonability of those. One way to help the reader would be to show the TS diagrams both upstream and downstream of the transformation region. Another would be to estimate the volume fluxes of incoming and transformed waters. For example, in Figure 2, the downstream water is of nearly the same density as the upper layer upstream. Is it reasonable that the upstream volume fluxes combine into this light water mass, or has the dense water simply not reached the downstream section yet? The same is the case in Fig. 4, and there I would like some further arguments that the sharp front is a stationary control rather than a prop- agating bore. An energy argumentation would also help. Is the loss of kinetic energy sufficient to explain the claimed rise in potential energy due to mixing? I think it is worth putting some effort into this, since this is one of the most extreme cases I have seen regarding mixing and removal of stratification downstream of a jump. That should also be one of the main key points of the manuscript.

AUTHORS' RESPONSE: We agree with the referee that the discussion - and the manuscript in general - is rather qualitative. Given the nature and the extent of the data material we are not able to provide a much more quantitative analysis without be- coming too speculative. At this stage in our research we have focused on showing the presence of internal hydraulic control in the Little Belt and documenting the conditions under which it occurs and the influence that it has on the lowest trophic levels of the ecosystem. However, the referee has made several suggestions as to how to make the manuscript more quantitative and informative, most of which we are able to follow. What we are unable to pursue is the energy argumentation, including the change of kinetic energy and the increase in potential energy due to mixing. A direct assessment of this would require concurrent, along-strait observations of flow velocity, which we

don't have. The referee has a very good point though, which is indeed worth putting effort into, we agree. When we have better, more detailed observations available we are going to pursue this idea.

AUTHORS' CHANGES IN MANUSCRIPT: We will revise Figs. 2 and 4 to show in detail how the mixing is taking place. This will be done by adding three TS diagrams that show the water masses in different locations (upstream, downstream and somewhere in between) along the two transects. This will allow us to determine the mixing ratios and to speculate on the mixing rates. We will revise the text accordingly. In addition, we will use the TS diagrams shown in Figs. 7 and 8 to discuss and compare the mixing ratios found in the two contrasting situations in which the upper or the lower layer become the active, accelerating one. In case of Fig. 2 this will also allow us to answer the referee's question concerning the fate of the mixed water masses on the downstream side.

COMMENT FROM REFEREE: 2. I miss a discussion about the importance of the curvature of the strait. The strait is dramatically meandering, and transects upstream of the most curving sections show strong interface tilting. How much can the interface be expected to tilt in the curving parts, and how does that influence the along-strait transects? May it also influence the hydraulics, the mixing, and the friction in the strait? AUTHORS' RESPONSE AND CHANGES IN MANUSCRIPT: The referee has a good point here. Curvature and secondary flows certainly play a big role in the narrow part of the Little Belt, but are of much lesser importance for the internal hydraulic control as such, which occur some distance upstream of the narrow part. We will revise the text to include a discussion and references on the possible effects of curvature and secondary flows.

COMMENT FROM REFEREE: 3. In the claimed scarcity of published observations of controlled flows the authors miss a number of high-quality observations over fjord sills that show the controlls and asso-ciated mixing in such flow configurations in much more detail than what is presented here, e.g. Farmer and Armi (1999), Klymak and

Gregg (2004), Inall et al. (2004), Staalstrom et al. (2015). AUTHORS' RESPONSE AND CHANGES IN MANUSCRIPT: We cannot explain why we have missed these references. We will carefully assess the information contained in these papers and will revise the text accordingly.

COMMENT FROM REFEREE: Detailed comments: Line 22: "that" should be "than". AUTHORS' RESPONSE AND CHANGES IN MANUSCRIPT: Yes. We will fix it.

COMMENT FROM REFEREE: Paragraph starting at line 48: Include references to high resolution fjord sill studies. AUTHORS' RESPONSE AND CHANGES IN MANUSCRIPT: Yes, cf. our reply above.

COMMENT FROM REFEREE: Lines 91-93: How much larger is the mixing efficiency in supercritical flow? I find no such conclusion in the referred paper. I would also like some references where the stratification breaks down totally after a control. Most of those I know about maintain a stratification, although modified, downstream of the control. AUTHORS' RESPONSE AND CHANGES IN MANUSCRIPT: Prastowo et al. (2009) report mixing efficiencies of around 0.11, which they compare with values of up to 0.20 reported in the literature. We don't find it necessary to revise the text, but will consider doing so to help the reader. We have no knowledge of observations of similar situations in which stratification breaks down completely. We will consider adding this information also.

COMMENT FROM REFEREE: Line 113: "reducing" should be "changing" since reducing is only applicable in one end. AUTHORS' RESPONSE AND CHANGES IN MANUSCRIPT: We don't understand what the referee means, but will try to word differently.

COMMENT FROM REFEREE: Line 126: "refraction" is the change of wave propagation direction due to change in propagation velocity, which is not what happens here. AUTHORS' RESPONSE AND CHANGES IN MANUSCRIPT: We agree. We will find a correct way of expressing what's taking place.

COMMENT FROM REFEREE: Line 142: "vertical profile" is misleading since the probe does not move vertically. AUTHORS' RESPONSE AND CHANGES IN MANUSCRIPT: We agree and will modify accordingly.

COMMENT FROM REFEREE: Line 168: "less and somewhat linearly stratified". Unclear. AUTHORS' RESPONSE AND CHANGES IN MANUSCRIPT: Yes. We will be more specific.

COMMENT FROM REFEREE: Figures 2 and 4: Show TS data both upstream and downstream. Also indicate in the transect where the TS diagrams have been taken. AUTHORS' RESPONSE AND CHANGES IN MANUSCRIPT: This follows from the referee's general comments. As we have explained above, we will modify Figs. 2 and 4.

COMMENT FROM REFEREE: Figure 3: Please indicate the timing of the various transects. AUTHORS' RESPONSE AND CHANGES IN MANUSCRIPT: We will add this information to Fig. 3.

COMMENT FROM REFEREE: Lines 208-210: How short are the length scales and how short should they be for the flow to be quasi-steady? AUTHORS' RESPONSE AND CHANGES IN MANUSCRIPT: The length scale to be considered is the distance over which the width changes, i.e., about 2 km. For the flow to be quasi-steady this should be well below with the distance travelled by a baroclinic wave (at a speed of about 1 m s-1) during a period equal to the time scale (about 12 hours), i.e. more than 10 km. We will try to clarify this in the text.

COMMENT FROM REFEREE: Lines 221-222: Please add a quantitative discussion about the reasonability of this. Is it sure that the front is not moving? AUTHORS' RESPONSE AND CHANGES IN MANUSCRIPT: A revision of Figure 4 should be able to help clarify this. In principal we cannot be sure that the front is not moving because we did not observe the front specifically. However, we did observe the same transect about 6 hours earlier, reported in Lund-Hansen et al. (2008), and found roughly the

same conditions. We also note that the salinity and temperature time series from the narrow part show that the stratification is often broken down, showing that the front isn't simply advected through the Little Belt. We will try to modify the text to clarify further.

COMMENT FROM REFEREE: Lines 229-230: What happens downstream when the curvature is much more impor-tant? AUTHORS' RESPONSE AND CHANGES IN MANUSCRIPT: This we will try to explain with reference to observations from Øre-sund, another Danish strait, reported in Nielsen (2001).

COMMENT FROM REFEREE: Lines 261-263: Strictly speaking what is shown is that the flow goes from subcritical to supercritical. You do not show where the critical cross-section is, and that it is steady in time. I.e. you do not show that the flow is controlled. AUTHORS' RESPONSE AND CHANGES IN MANUSCRIPT: We believe that we have shown that the flow can be considered quasi-steady and so that the point of control must be located somewhere between the two stations. Under different flow conditions the point of control could be located elsewhere though. We will put more emphasis on discussing the variability of the flow conditions and the effects thereof with respect to hydraulic control.

COMMENT FROM REFEREE: Lines 292-295: Again, argue more quantitatively. The evidence of a control is quite weak. AUTHORS' RESPONSE AND CHANGES IN MANUSCRIPT: We admit that the argumentation could be improved here. We will argue more quantitatively using the mixing ratios, as described above, and we will re-verse the order of the arguments roughly as follows. The flow speed observed in the narrow part (around or above 1 m s-1) along with the weakened stratification shows that the flow is supercritical here. Since the conditions on the upstream side are certainly subcritical, a point of control must exist and be located somewhere in the contracting part to the north. The structure of the water column observed on the upstream side shows that the upper layer is the active, accelerating one, which is entraining water from below. This is supported by the water masses observed in the narrow part of the Little Belt.

COMMENT FROM REFEREE: Lines 314-320: How does the increased primary production in the upper layer fit with the earlier conclusion that entrainment is mainly into the bottom layer? AUTHORS' RESPONSE AND CHANGES IN MANUSCRIPT: When the intensely mixed water masses are advected away from the narrow part of the Little Belt, they are brought upward in the water column and into the photic zone and are spreading sideways. This is simply due to the shallow depth on the downstream side. Fig. 2 should be able to show this. We will modify the text to clarify this.

COMMENT FROM REFEREE: Lines 326-327: Again this is somewhat confusing. Entrainment is mainly into the upper layer, but downstream the upper layer is at the bottom. Some more explanation needed to make the text clear. AUTHORS' RESPONSE AND CHANGES IN MANUSCRIPT: The density of the mixed water masses and the stratification found on the downstream side determine where in the water column the mixed water masses end up. Even if this is relatively deep in the water column, increased phytoplankton concentration may be the result still. We shall explain better.

COMMENT FROM REFEREE: Line 340: In estuarine flows the exchange is often determined by internal hydraulics rather than sea level differences. AUTHORS' RESPONSE AND CHANGES IN MANUSCRIPT: Yes, but a great deal of estuarine examples exist in which there is a substantial barotropic pressure gradient that may drive both layers in the same direction. We will try to be more specific about the situations that we are considering.

COMMENT FROM REFEREE: Lines 371-372: Again a quantitative discussion is needed. How quickly is Little Belt flushed out? AUTHORS' RESPONSE AND CHANGES IN MANUSCRIPT: Yes, this piece of information is needed. We will add it both here and in the section on the Physical Setting.

COMMENT FROM REFEREE: Lines 381-383: How can information about the critical point be used to quantify mixing? AUTHORS' RESPONSE AND CHANGES IN MANUSCRIPT: Given a 'weir formula', cf. Pratt (2004), the production of kinetic energy and a mixing efficiency it should ideally be possibly to quantify mixing. We will revise the text to explain this.

COMMENT FROM REFEREE: Line 412: Since such phenomena have not been observed, one could include "the possibility" to avoid stating that they exist in reality without evidence. AUTHORS' RESPONSE AND CHANGES IN MANUSCRIPT: That is true. We will follow the referee's recommendation.

---

## Author Comment (AC2) · 12 Jul 2017

COMMENTS FROM REFEREE: The manuscript "Internal hydraulic control in the Little Belt, Denmark. Observations of flow configurations and water mass formation" by Nielsen et al. investigates obser-vations in the northern part of the Little Belt during two-layered, southward flow and connect them to hydraulic control. Two configurations are described, where either the upper or the lower layer is the active and accelerating one. These configurations are dependent on the depth of the pycnocline on the upstream side. The finding of this study are interesting and worth publishing, after following comments are addressed. Line 25: Although many readers may be familiar

with the concept of hydraulic con-trol, it would be useful to give a short definition (with reference) of this concept in the introduction as it is the main topic of this study. AUTHORS' RESPONSE AND CHANGES IN MANUSCRIPT: We will add a definition and list important aspects of hydraulic control with suitable references.

COMMENTS FROM REFEREE: Line 48-49: please include references to these studies. AUTHORS' RESPONSE AND CHANGES IN MANUSCRIPT: These reference are listed in the paragraph in question (lines 48-64). But since this is unclear, we shall clarify.

COMMENTS FROM REFEREE: Line 119ff: Baltic→Baltic Sea (several times) AUTHORS' RESPONSE AND CHANGES IN MANUSCRIPT: Instead of 'Baltic' we shall write 'Baltic Sea'.

COMMENTS FROM REFEREE: Method section: It would be great to have all observations used in this study shortly described in form of a table including information about location (refer to figure 1 using the station/transect numbers), the observation period, what was observed and how, and important notes (e.g. that there was an inflow during the observation period). AUTHORS' RESPONSE AND CHANGES IN MANUSCRIPT: This is a good point. We believe that we can briefly describe all the relevant information concerning the observations by modifying the text at the beginning of the Methods section. By adding the timing of the different observations to Fig. 3 the reader can get an overview that a table would otherwise provide.

COMMENTS FROM REFEREE: The name of section 3 (Methods) should be renamed to "Observations" and the name of section 4 (Observation and discussion) to "Results" AUTHORS' RESPONSE AND CHANGES IN MANUSCRIPT: Section 3 is concerned with instrumentation and how data were collected and handled, not observations. Section 4 contains both observations and discussion pertaining to these. Section 5 contains a more general discussion. Therefore we do not agree with the referee and would like to keep the section headings.

COMMENTS FROM REFEREE: How do other processes, for example wind mixing, may have impacted the observed signals and their interpretation? AUTHORS' RE-SPONSE AND CHANGES IN MANUSCRIPT: Under most circumstances mixing due to hydraulic control is going to be the most important mixing agent by far. We will add a brief discussion on this question, including situations where wind mixing is actually the more important one.

COMMENTS FROM REFEREE: Line 205-206: please explain in more detail why effects, such as mixing can be ruled out. AUTHORS' RESPONSE AND CHANGES IN MANUSCRIPT: Yes, this could probably use a bit of explanation. We will explain and clarify.

COMMENTS FROM REFEREE: Line 209: please quantify "short enough" AUTHORS' RESPONSE AND CHANGES IN MANUSCRIPT: The length scale to be considered is the distance over which the width changes, which is about 2 km in this case. For the flow to be quasi-steady, this length scale should be well below the distance travelled by a baroclinic wave (at a speed of about 1 m s-1) during a period equal to the time scale (about 12 hours), i.e. more than 10 km. We will try to clarify this in the text.

COMMENTS FROM REFEREE: Line 229: please discuss the "considerable adjustment" in more detailed AUTHORS' RESPONSE AND CHANGES IN MANUSCRIPT: Despite the cross-strait variability in the two transects (due to geostrophy), the observations clearly show a deepening of the interface. We shall clarify this in the text.

COMMENTS FROM REFEREE: Line 233ff: the definition of the two layer system in the cross-strait transects in Figure 5 is somehow unclear. Which case, described here, is assigned to the situation in Figure 5? Please clarify. AUTHORS' RESPONSE AND CHANGES IN MANUSCRIPT: That may be a good point. We shall modify the text to clarify this.

COMMENTS FROM REFEREE: Line 260ff: The computed Froude numbers indicate a change from subcritical to su-percritical flow. Please discuss in more detail how this

finding gives evidence that the situation is subject to hydraulic control. AUTHORS' RESPONSE AND CHANGES IN MANUSCRIPT: If the flow is quasi-steady then a change from sub to supercritical flow implies that the flow passes a point that is critical, at which the flow is subject to control. The definition of hydraulic control to be added to the introduction should help to understand this, but we will clarify further at this point in the text.

COMMENTS FROM REFEREE: Line 267-268: How is the location of the point of control be influenced by fric-tion/entrainment? AUTHORS' RESPONSE AND CHANGES IN MANUSCRIPT: Friction and/or entrainment shift the point of control away from the top of the slope. This is explained in detail in the two references, and so further explanation should not be necessary.

COMMENTS FROM REFEREE: Line 310ff: The description of the distributions of chlorophyll a is somehow confusing. Please first describe Figure 7 and then Figure 8. AUTHORS' RESPONSE AND CHANGES IN MANUSCRIPT: With respect to the chlorophyll concentration, Figs. 7 and 8 show essentially the same picture. So describing first Fig. 7 and then Fig. 8 would imply repeating the description, which we should avoid. However, we will try to modify the text to express more clearly what is observed.

COMMENTS FROM REFEREE: Line 381: How can hydraulic control be used to quantify the transport of water masses, the mixing between them and also the influence of other processes? AUTHORS' RESPONSE AND CHANGES IN MANUSCRIPT: Given a 'weir formula', cf. Pratt (2004), the production of kinetic energy and a mixing efficiency it should ideally be possibly to quantify mixing. We will revise the text to explain this.

COMMENTS FROM REFEREE: Figure 1: To more easily identify the position of the stations, it would be great if the station positions could be more highlighted and the transects should be given a num-bering. The depth contours are difficult to distinguish.

AUTHORS' RESPONSE AND CHANGES IN MANUSCRIPT: To better show the locations of the stations we will change the symbol of these. It shouldn't be necessary to number the transects since there are only two and they appear in separate parts of the figure. When printed or scaled on the computer screen (which is probably what most readers do) the figure becomes much easier to read.

COMMENTS FROM REFEREE: Figure 2: In the title it is written that the length of the transect being measured from the northern end (assuming distance=0), but looking at the figure itself and the text, it looks like the approximately two layered water masses are located at 50 km distance. Please include in the figure, where north and south is or the starting/ending coordinates. AUTHORS' RESPONSE AND CHANGES IN MANUSCRIPT: We are somewhat confused here. Maybe the referee misunderstood something. In Fig. 2 at 50 km there is only one contour line of density, showing that the water column is almost fully mixed. So, clearly the strongly stratified water masses are near 0 km. We will be modifying Figs. 2 and 4 to show detailed TS diagrams. This should limit any further confusion.

COMMENTS FROM REFEREE: Figure 3 The different curves are difficult to distinguish and need to be clarified AUTHORS' RESPONSE AND CHANGES IN MANUSCRIPT: We agree that the different curves in Fig. 3 are difficult to distinguish. However, we would like to avoid adding more details as this would clutter the figure even more. Also, we would like to avoid adding colors. However, we will add a short description on how to better read the figure. Notably, when the curves are nearly coinciding and thus most difficult to distinguish, it implies that there is no velocity shear or stratification.

COMMENTS FROM REFEREE: All figures showing salinity: The unit is missing everywhere, is it g/kg? AUTHORS' RESPONSE AND CHANGES IN MANUSCRIPT: We are using Practical Salinity Units (dimensionless) everywhere, cf. our reference to UNESCO (1981). We will be more specific about this in the figures and throughout the text.

---

## Author Comment (AC3) · 12 Jul 2017

COMMENTS FROM REFEREE: This paper on internal hydraulic control in the Little Belt by Nielsen et al. is of consider-able interest. The introduction comprises an overview of the present state of multi-layer hydraulics and sections 2 and 3 (physical setting, methods) provide adequate descrip-tions of the location and fieldwork. My main comments pertain to section 4 dealing with the observations and discussion. First some minor points: line 179: generally the expression used is "back and forth", but the authors may possi-bly have sopme ingeious reason for reversing the standard order. AUTHORS' RESPONSE AND CHANGES IN MANUSCRIPT: We have no particular

reason for using this order. If "back and forth" is avoiding confusion then we shall modify the text.

COMMENTS FROM REFEREE: line 181: pycnocline (singular) is located AUTHORS' RESPONSE AND CHANGES IN MANUSCRIPT: We will fix the typo.

COMMENTS FROM REFEREE: line 199: for heightened clarity write ... the upper and lower water masses of the Kattegat with a density difference... AUTHORS' RESPONSE AND CHANGES IN MANUSCRIPT: That is a good suggestion, which we will follow.

COMMENTS FROM REFEREE: These were minor matters but my next comment is more important: line 228: It is stated that the cross-strait density transects showed density variations across the strait, cf. fig. 5. An examination of in particular the right panel of fig 5 reveals a slope of such a magnitude that a geostrophic balance is a distinct possibility, and hence that it is doubtful whether the system can be regarded as nonrotational and farther down the page line 255 et seq. the authors apply Armi's nonrotating formalism. It is necessary to include a discussion of why rotational effects can be neglected!! AUTHORS' RESPONSE AND CHANGES IN MANUSCRIPT: Yes, it is true that rotational effects cannot be neglected fully, even if much can be learned based on Armi's non-rotating formalism. We will modify the text and discuss this.

COMMENTS FROM REFEREE: Sections 5 and 6 (discussion and conclusion) are satisfactory and I particularly appre-ciated that the authors invoked the hydraulic effects on the biogeochemical processes, something I do not think has been done before. A minor point here is that on line 390 ... internal dynamics (plural) are not. . AUTHORS' RESPONSE AND CHANGES IN MANUSCRIPT: We will fix the typo.

---

## Author Response (AR2)

Reply to Anonymous Referee #1

COMMENTS FROM REFEREE: I am happy with the modifications that give me more confidence that we are actually seeing quasi-stationary hydraulic controlls. However, it could be taken one step further by just considering mass conservation. In a steady system with fully mixed downstream water, the downstream salinity must be determined by the salinities and relative volume fluxes of the two upstream layers (e.g Arneborg, 2016, JGR, 121, 2035–2040). That could be mentioned at some parts of the text, as suggested in the detailed comments below. I also think that the discussion about Chl_a and primary production could be extended somewhat, as also specified in more detail below.

Detailed comments:
Line 96-97: I still don't understand where this comes from. Prastewo et al (2009) find a mixing efficiency of about 0.11 which is less than the standard value of 0.2 used by most other researchers. Mixing may be small in sub-critical flow, but there is no reason to believe that the mixing efficiency is 10 times smaller than 0.11. In flows dominated by bottom friction, the bulk mixing efficiency may be smaller since most of the turbulent dissipation happens in the well mixed bottom boundary layer, but that is not really what you write.
AUTHORS' RESPONSE AND CHANGES IN MANUSCRIPT: We are having difficulties understanding what the referee finds to be a problem here. Our point is that mixing is much more intense in the supercritical part of the flow compared to the upstream, subcritical part. Existing studies have found the difference in mixing efficiency to be an order of magnitude, i.e. O(0.1) for supercritical flows and O(0.01) for subcritical flows. Is the referee concerned with the exact values of the efficiencies? We are deliberately not going into details with these because they are not so important. Therefore we are unable to see how our text is unclear about the point that we are trying to make. However, given the referee's criticism we are going to try to reformulate the two sentences in question.

COMMENTS FROM REFEREE: Line 186: Is it reasonably to assume that 80% of the volume flux takes place in the upper layer, as these results indicate? I think it is, but I also think this extra constraint on the results should be mentioned and evaluated, even though the evaluation cannot be very precise without velocity measurements.
AUTHORS' RESPONSE AND CHANGES IN MANUSCRIPT: This is a very good point that deserves attention. We will add both a general comment earlier in the manuscript, including a reference to Arneborg (2016), and a specific comment in the following paragraph.

COMMENTS FROM REFEREE: Lines 198-201: To me this is not that clear to see directly from Figs 1 and 2, but it may be an interpretation of the results. Please write some more explanation.
AUTHORS' RESPONSE AND CHANGES IN MANUSCRIPT: We will add some details to clarify this.

COMMENTS FROM REFEREE: Lines 234-236: Same as above. Is it reasonable to assume that 60% of the volume flux takes place in the upper layer?
AUTHORS' RESPONSE AND CHANGES IN MANUSCRIPT: This is a good point and question. We will add some text to answer it.

COMMENTS FROM REFEREE: Line 251: I had some problems understanding this. Maybe add "of the strait" to make clear what "upper part" refers to.
AUTHORS' RESPONSE AND CHANGES IN MANUSCRIPT: We agree and will modify the text.

COMMENTS FROM REFEREE: Line 255: Change to "…top of the slope."

AUTHORS' RESPONSE AND CHANGES IN MANUSCRIPT: A typo. We will fix it.

COMMENTS FROM REFEREE: Line 256: Change to "…in case of a relatively…"
AUTHORS' RESPONSE AND CHANGES IN MANUSCRIPT: Another typo. which we will fix.

COMMENTS FROM REFEREE: Line 258: Change to "In nature, hydraulic controls…"
AUTHORS' RESPONSE AND CHANGES IN MANUSCRIPT: Yes, a comma is needed, which we will add.

COMMENTS FROM REFEREE: Lines 359-362: I understand this explanation for the upper layer in Fig. 8, where bottom water is mixed up into the upper layer. However, I would like some more text on how the authors interpret the enhanced Chl_a in the lower layers. Why does entrainment of upper layer water enhance primary production in the lower layer? The nutrients are already there and the light is probably not better than upstream. Is it because primary producing plankton are mixed down into the layer where they can grow on the limited light sources there?
AUTHORS' RESPONSE AND CHANGES IN MANUSCRIPT: Yes, this is not clear. We will clarify the text.

COMMENTS FROM REFEREE: Lines 404-408: In my way of seing this, it is the control that determines the downstream salinity through the relative volume fluxes in the two layers. The entrainment can do whatever it wants as long as it is strong enough to homogenize the fluid.
AUTHORS' RESPONSE AND CHANGES IN MANUSCRIPT: Yes, this is true. But our concern here is how the flow rates, and subsequently the water masses on the downstream side, are determined by the control. We see no reason to modify this part of the discussion.

COMMENTS FROM REFEREE: Line 437: Change to "..downstream of the sill."
AUTHORS' RESPONSE AND CHANGES IN MANUSCRIPT: Yes, a typo, which we will fix.

COMMENTS FROM REFEREE: Lines 477-478: This follows from the control limiting mainly the volume flux in the upper, active, layer.
AUTHORS' RESPONSE AND CHANGES IN MANUSCRIPT: Yes. We believe that we have explained this adequately and see no reason to modify the text.

[revised manuscript text omitted]
. In supercritical flows the efficiency of mixing between the water masses is an order of magnitude higher than in subcritical flows (Prastowo et al., 2009, and references therein). This is due to growing instabilities at the pycnocline (Pawlak and Armi, 2000). This implies that downstream of the point of hydraulic control intense mixing is taking place, causing the formation of new water masses that are distinctly different from those on the upstream side. The conditions at the point of control have a strong bearing on the properties of these new water masses. For reasons of continuity, if the mixing in the supercritical part of the flow is strong enough to make the water masses completely homogeneous, the salinity on the downstream side is directly determined by the salinities of the water masses on the upstream side and the ratio of the flow rates at the control (e.g. Arneborg, 2016). 
[revised manuscript text omitted]
. Under these circumstances the salinity in the well-mixed water masses on the downstream side indicates that the hydraulic control allows for an inflow of 80 % upper layer water and 20 % lower layer water from the upstream side. Given the deep location of the pycnocline, this would seem a probable result. We have previously argued that this rapid adjustment is connected with hydraulic control and that the intense mixing of the water masses is a result thereof (Lund-Hansen et al., 2008). The observations shown in Fig. 2 are not providing new information in this respect. However, the observations to be shown in the following will shed new light on hydraulic control in this area, including details of the adjustment and the effects on the formation of water masses. What Figs. 1 and 2 do show clearly though is that when well-mixed water masses are exported from the deep, narrow part of the Little Belt, covering a depth of about 50 m and a width of about 1 km, they are brought upward in the water column and are able to spread over a wide, shallow area, less than 20 m deep and several kilometres wide. Even though the mixed water masses could still be confined relatively deeply in the water column, this upwelling could well drive an increased growth of phytoplankton (Lyngsgaard et al., 2014).

Figure 3 shows time series of velocity, salinity and temperature obtained at the fixed station at the Old Bridge (see Fig. 1).

The time series cover the two periods that include the observations shown in Figs. 4 through 8, the timings of these observations being indicated. The observations show the conditions that often prevail in the narrow part of the Little Belt. There is a strong tidal signal of diurnal frequency and amplitude ranging from the 0.5 to 1 m s$^{-1}$. In addition, there is a flow component that varies on a longer time scale, of a few days or so, which is primarily caused by wind-generated water level variations in the Kattegat and in the western Baltic Sea (Jacobsen, 1980). This flow is of a considerable magnitude as well and may reach values of 1 m s$^{-1}$ or more. Figure 3 also shows that the water masses in the narrow part of the Little Belt are usually weakly stratified, despite the strong stratification often found at least in the Kattegat to the north (Nielsen, 2005). Given the magnitude of the barotropic flow we can quickly deduce that the flow in the narrow part of the Little Belt is often supercritical with respect to all internal modes. This circumstance was also pointed out by Jakobsen and Ottavi (1997). This is so even for the strongest stratification that one can imagine, which is a two-layer structure corresponding to the upper and lower layer water masses of the Kattegat of a density difference of about 14 kg m$^{-3}$ (Nielsen, 2005). In a 30 m water column of layers of equal thickness this corresponds to a long-wave speed of about 1 m s$^{-1}$ (e.g., Armi, 1986). As the cross-sectional area increases in either direction we can also deduce that situations of supercritical flow in the narrow part are associated with subcritical flow on the upstream side and so that the flow must often be subject to hydraulic control, the control point to be found at the upstream contracting part of the Little Belt. We may therefore conclude that our assumptions so far about the observations of hydraulically controlled flows have been correct (Lund-Hansen et al., 2008). Further, we may point out that this finding is significantly above measurement errors. Thus, various effects that could dampen the flow below criticality, such as friction or mixing, are seen to be of less importance (Pratt,1986; Nielsen et al., 2004). On the other hand, hydraulic control in the Little Belt is an intermittent phenomenon that is limited to parts of the tidal cycle if the barotropic flow is weak, which is the case in some of the periods shown in Fig. 3. The length scales of the contracting parts, about 2 km, are short compared to the distance travelled by a baroclinic wave during a tidal period (more than 10 km). This implies that the hydraulic adjustment is well toward the quasi-steady limit (Helfrich, 1995).

Another along-strait transect, obtained using the ScanFish as described above, is shown in Fig. 4. It covers the contracting part to the north and some of the narrow part, as shown by the line and circles in the detailed inset in Fig. 1. The transect was made on 16 June 2004 from 12:22 to 13:37 at the time of which the flow was southward through the Little Belt, the depth-averaged flow being well above 1 m s$^{-1}$ for a period of about 24 hours, cf. Fig. 3. At the upstream end of the transect the water column is approximately two-layered, the salinity and the temperature being about 22 and 13.5 °C in the upper layer and 29 and 7.5 °C in the lower layer, cf. panel (b) in Fig. 4. Toward the narrow part of the Little Belt the water column is undergoing rapid adjustment, the pycnocline dropping toward the bottom from a depth of less than 10 m over a distance of a few kilometres, and the lower layer being subject to rapid dilution with upper layer water, cf. Figure 4, panel (c). In the narrow part of the strait the water masses are almost well-mixed, the mixing ratio being around 60 % upper layer water and 40 % lower layer water compared to the upstream side, cf. Figure 4, panel (d). From these circumstances we may safely conclude that the flow is subject to hydraulic control and that the lower layer is the active one. Furthermore, since the water column becomes fully mixed on the downstream side we may conclude that the hydraulic control allows a ratio of flow rates of the inflowing water masses equal to the mixing ratio. Due to shallower depth of the pycnocline in this situation 
[revised manuscript text omitted]
 from the lower layer together with nutrient-depleted phytoplankton from the upper layer, the result is a quick response  
[revised manuscript text omitted]